# Identification of the Axis β-Catenin–BTK in the Dynamic Adhesion of Chronic Lymphocytic Leukemia Cells to Their Microenvironment

**DOI:** 10.3390/ijms242417623

**Published:** 2023-12-18

**Authors:** Imane Mihoub, Tareck Rharass, Souhaïl Ouriemmi, Antonin Oudar, Laure Aubard, Valérie Gratio, Gregory Lazarian, Jordan Ferreira, Elisabetta Dondi, Florence Cymbalista, Vincent Levy, Fanny Baran-Marszak, Nadine Varin-Blank, Dominique Ledoux, Christine Le Roy, Laura Gardano

**Affiliations:** 1INSERM, U978, 93000 Bobigny, France; mihoub.imane993@gmail.com (I.M.); souhail.ouriemmi@univ-paris13.fr (S.O.); antonin.oudar@inserm.fr (A.O.); laure.aubard@inserm.fr (L.A.); gregory.lazarian@aphp.fr (G.L.); jordan.ferreira@univ-paris13.fr (J.F.); elisabetta.dondi@inserm.fr (E.D.); florence.cymbalista@aphp.fr (F.C.); fanny.baran-marszak@aphp.fr (F.B.-M.); ledoux@univ-paris13.fr (D.L.); christine.le-roy@inserm.fr (C.L.R.); 2UFR SMBH, LabEx INFLAMEX, Université Paris 13—«Sorbonne Paris Nord», 93000 Bobigny, France; 3INSERM U1149, Université Paris Cité, Hôpital Bichat, 75018 Paris, France; valerie.gratio@inserm.fr; 4AP-HP Hôpital Avicenne, 93000 Bobigny, France; 5URC, AP-HP Hôpital Avicenne, 93000 Bobigny, France; vincent.levy@aphp.fr

**Keywords:** β-catenin, microenvironment, CLL, BTK, stromal cells

## Abstract

In the microenvironment, cell interactions are established between different cell types to regulate their migration, survival and activation. β-Catenin is a multifunctional protein that stabilizes cell–cell interactions and regulates cell survival through its transcriptional activity. We used chronic lymphocytic leukemia (CLL) cells as a cellular model to study the role of β-catenin in regulating the adhesion of tumor cells to their microenvironment, which is necessary for tumor cell survival and accumulation. When co-cultured with a stromal cell line (HS-5), a fraction of the CLL cells adhere to stromal cells in a dynamic fashion regulated by the different levels of β-catenin expression. In non-adherent cells, β-catenin is stabilized in the cytosol and translocates into the nucleus, increasing the expression of cyclin D1. In adherent cells, the level of cytosolic β-catenin is low but membrane β-catenin helps to stabilize the adhesion of CLL to stromal cells. Indeed, the overexpression of β-catenin enhances the interaction of CLL with HS-5 cells, suggesting that this protein behaves as a regulator of cell adhesion to the stromal component and of the transcriptional regulation of cell survival. Inhibitors that block the stabilization of β-catenin alter this equilibrium and effectively disrupt the support that CLL cells receive from the cross-talk with the stroma.

## 1. Introduction

The trafficking of immune cells between the blood and secondary lymphoid organs is the result of the interplay between concentrations of specific chemokines and the expression of their receptors on immune cells. In the microenvironment, e.g., in blood, bone marrow or lymph nodes, immune cells interact with the different cell populations and regulate their trafficking, i.e., the balance between adhesion and migration [1]. The interaction with the microenvironment is critical for the tissue-specific accumulation of tumor cells in the majority of lymphomas, including mantle cell lymphoma (MCL) and chronic lymphocytic leukemia (CLL), among others [2]. MCL and CLL cells rapidly undergo apoptosis when isolated from a patients’ blood, whereas culture them on a feeder layer prolongs their life and allows them to be maintained in culture for a long period of time [3]. This feeder layer, composed of bone-marrow-derived stromal cells, is used to mimic the microenvironment and study the interaction established between tumor cells and the stromal component of the microenvironment. Among the plethora of signaling pathways that are activated by the microenvironment and elicit a survival response, the Wnt signaling pathway has received much attention in CLL in recent years, as Wnt signaling effectors are frequently mutated in CLL cells, although none are drivers of the disease [4,5,6,7]. However, mutations that disable genes encoding Wnt signaling effectors, including β-catenin, exacerbate CLL phenotypes, suggesting that this pathway plays an important role in the hallmarks of the disease [6]. Wnt refers to a family of 19 glycoproteins that interact with cellular receptors to initiate signaling cascades to control cell proliferation, migration and differentiation [8]. The Wnt signaling pathway can be divided into canonical and non-canonical signaling pathways based on their dependence on the protein β-catenin as a signaling effector [8]. The non-canonical signaling pathway involves membrane proteins that mainly control the level of planar cell polarity (PCP) [9]. In CLL, the ligands Wnt5A and Wnt5B, which regulate the non-canonical Wnt/PCP pathway, and their effectors are overexpressed, and they regulate the communication of CLL cells with the microenvironment [10]. In addition, the activation of Wnt5a signaling enhances CLL chemotaxis towards CXCL12 [11,12,13]. On the other hand, the canonical signaling pathway depends on β-catenin, which is stabilized in response to Wnt ligands and translocates into the nucleus where it participates in the transcription of genes involved in cell cycle regulation, proliferation and migration [14]. In the absence of Wnt, β-catenin is degraded by the proteasome following its phosphorylation by casein kinase I at the serine residue S45, followed by GSK3β at S33, S37 and T41. According to the model, Wnt ligands inhibit GSK3β activity, thereby reducing the phosphorylation and degradation of β-catenin [15]. Upon its translocation into the nucleus, β-catenin interacts with the TCF/LEF family of transcription factors and activates the transcription of Wnt-dependent target genes. However, the interaction of β-catenin with nuclear partners is not restricted to the TCF/LEF family, as other transcription factors, such as Sox, FOXO and NF-κB are known to interact with β-catenin via both Wnt-dependent and -independent routes [16]. Previously, we have shown that β-catenin is a BCR effector in MCL, as it is stabilized in a BTK-dependent manner upon antigenic stimulation [17]. Its translocation into the nucleus contributes to the transcription of NF-κB target genes such as IL-6, thereby increasing the survival of MCL cells. In parallel to its involvement in the Wnt signaling pathway, β-catenin has other functions, such as stabilizing cellular junctions by interacting with the intracellular domain of cadherins [18]. The interaction between β-catenin and cadherins has been described to have both positive and negative roles on its transcriptional behavior, depending on the cell type [14,19,20]. Cadherins mediate homo- and heterotypic intercellular interactions, although they can also affect signaling pathways where ligands and receptors are exposed on the cell surfaces of interacting cells, e.g., notch/delta [21]. In a co-culture model of CLL cells with bone marrow stromal cells, stromal Notch2 was shown to contribute to β-catenin stabilization in CLL cells to enhance survival signals in the microenvironment [22]. In CLL cells, the kinase BTK integrates signals from the microenvironment as downstream effectors of several chemokines, such as CXCL12 [23]. The stimulation of CXCR4 by its ligand CXCL12 leads to the very rapid phosphorylation of BTK, followed by integrin activation, which increases CLL cell migration and adhesion to VCAM-1 expressed on the surfaces of microenvironmental cells [24]. The activity of BTK is controlled by its phosphorylation status, with the Lyn-dependent phosphorylation of the tyrosine residue located at the position 551 in the C-terminus (Y551) priming its autophosphorylation at Y223 in the SH3 domain, which fully activates the catalytic activity of BTK [25]. However, BTK kinase activity is not required for all of its functions, suggesting that BTK also acts as an adaptor molecule [26].

As two signaling effectors from the microenvironment, we investigated the roles of β-catenin and BTK in CLL cells in a co-culture model with human stromal cells (HS-5). We describe a dynamic interaction of CLL cells with the stromal component, resulting in an adhesion–detachment equilibrium. In CLL cells non-adhering to stromal cells, β-catenin is upregulated and accumulates in the nucleus. Conversely, in adherent CLL cells to the stromal component, β-catenin is detected only at the cell membrane, suggesting a role in cell adhesion. This model is supported by the observation that the overexpression of β-catenin in CLL cells increases their adhesion to the stromal component. Thus, β-catenin represents an important element that regulates the adhesion of CLL cells to the microenvironment.

## 2. Results

### 2.1. Relationship between the Adhesion and Survival of CLL B Cells in the Microenvironment

CLL cells rely on their interaction with the microenvironment for their survival and tumor maintenance. To dissect the molecular basis of tumor cell–microenvironment communication, we used a validated co-culture system to reproduce in vitro the microenvironment in which CLL cells can survive [27]. The co-culturing of CLL B cells with a feeder layer of a bone marrow stromal cell line (HS-5) prolongs their survival, as shown by the increase in the percentage of viable cells when co-cultured for up to 6 days as compared to cells maintained in culture without a feeder layer for the same amount of time (Figure 1A). When tumor CLL B cells are cultured with HS-5 cells, a fraction of them promptly adhere to the feeder layer. When we compared the viability of cells recovered with stromal cells versus the cells in the medium, we found a higher percentage of viable cells associated with stromal cells, indicating the existence of a relationship between adhesion and survival (Figure 1B). To investigate this relationship, we first analyzed the dynamic of the interaction of the CLL cells with the stromal component. When Hg-3 cells, a CLL cell line, were co-cultured with HS-5 cells for different time points, we observed that the percentage and the number cells of Hg-3 associated with HS-5 cells after the removal of the non-adherent CLL cells did not significantly change over this time range (Appendix A). The same results were found with primary CLL cells co-cultured with HS-5 for up to 6 days (Figure 1C). Next, we asked whether CLL cells were able to detach from HS-5 cells so that the percentage of adherent cells was the result of a dynamic attachment–detachment process. Hg-3 cells were labeled with a cell tracker and allowed to adhere to a layer of HS-5 cells for 2 h. After the removal of the labeled non-adherent Hg-3 (green cells in Figure 1), an equal number of unlabeled Hg-3 cells (gray cells) was added and the percentages of labeled Hg-3 cells were analyzed in the adherent and non-adherent fractions (Figure 1D, left panel). Labeled Hg-3 cells were detected in both fractions, indicating that Hg-3 cells detach from HS-5 cells (Figure 1D, right panel). In addition, after 2 h of co-culture of Hg-3 cells with HS-5 cells, the non-adherent Hg-3 cells were transferred to a new feeder layer (referred to as primed Hg-3 cells) and allowed to adhere for a further 2 h. The ratio of the number of Hg-3 cells to HS-5 cells was compared in a co-culture of HS-5-primed Hg-3 cells versus non-primed cells and was found to be unchanged in both conditions, indicating that non-adherent cells still possess adhesive properties (Figure 1E).

Taken together, these results suggest that tumor B cell adhesion to HS-5 stromal culture is a dynamic process in which cells adhere and detach, probably in response to soluble factors but also to mechanical cues from the microenvironment that contribute to their survival.

### 2.2. β-Catenin Levels and Cellular Localization Differ between CLL Cells Adhering or Not to Stromal Cells

To investigate the link between adhesion and survival, we turned to β-catenin, a multifunctional protein involved in both the stabilization of adhesion complexes and the transcription of survival genes in several cell types.

An analysis of the dynamics of β-catenin in primary CLL cells not adherent to stromal cells was performed via Western blotting in co-cultures for different time periods. The level of β-catenin increased after 15 min of co-culture, reached maximum stability at 30 min and persisted for up to 2 h (Figure 2A). In addition to primary cells, we also observed β-catenin stabilization in the Hg-3 cells and Mec-1, another CLL cell line (Figure 2B). In order to confirm that the observed increase in β-catenin was only coming from B cells and not from contaminating stromal cells, we performed the kinetics with a conditioned medium (CM) issued from a 24 h culture of HS-5. We observed similar stabilization of β-catenin as in the presence of stromal cells (Figure 2C). These results indicate that soluble factors in the HS-5 supernatant triggered β-catenin stabilization in the CLL cells. Since β-catenin has different roles depending on its cellular localization, i.e., promoting adhesion when located at the plasma membrane or regulating transcription when in the nucleus, we determined whether β-catenin could play a role in regulating the balance between adherent and non-adherent cell behavior. We used flow cytometry to analyze the β-catenin levels in co-cultures distinguishing HS-5 and CLL cells labeled with anti-CD90 and -CD19 antibodies, respectively. Interestingly, β-catenin was detected at a lower percentage in adherent Mec-1 cells compared to Mec-1 cells that did not adhere to stromal cells (Figure 2D). These results were reproduced in Hg-3 cells (Appendix A). In addition, an analysis of β-catenin localization via imaging cytometry in adherent versus non-adherent cells revealed that a higher percentage of cells with nuclear β-catenin was present in the non-adherent cell population (Figure 2E).

The different levels of β-catenin in the two cell pools suggest two different signaling situations. In adherent cells, as in the absence of Wnt or other stimulation, the cytosolic β-catenin is constantly degraded, leaving intact only the pool of β-catenin localized at the cell membrane; in non-adherent cells, β-catenin is stabilized, probably in response to a soluble ligand, and translocates into the nucleus to regulate transcription.

### 2.3. BTK and β-Catenin Impact Cell Morphology in CLL Cells Co-Cultured with Stromal Cells

To test this hypothesis at a functional level, we first used confocal microscopy. CLL B cells in co-culture with HS-5 cells showed that adherent CLL cells form protrusions or ruffles to contact stromal cells, as evidenced by fluorescent phalloidin, which labels actin filaments, together with anti-CD20 to distinguish primary CLL from HS-5 cells (Figure 3A). Second, we analyzed via imaging cytometry cell doublets containing one Mec-1 cell in contact with one HS-5 cell and observed the localization of β-catenin at the contact surface between the two cell types. This polarized localization of β-catenin was observed despite the different levels of the protein in the two cell types (Figure 3B). These results were replicated in doublets containing Hg-3 and HS-5 cells in co-culture in the same conditions as for Mec-1 (Appendix A).

The protein kinase BTK, which is a downstream effector of BCR activation and of several chemokines, affects cell adhesion, migration and membrane dynamics, inducing the formation of membrane ruffles in various cell types through the activation of Rac and Cdc42 [28]. β-Catenin is known to regulate cell adhesion by bridging cadherins to the actin cytoskeleton but also localizes in the ruffles to regulate cell migration [20]. When we analyzed BTK and β-catenin in Mec-1 cells after two hours of co-culture, we observed that the adherent cells formed protrusions containing BTK and β-catenin, whereas the non-adherent cells maintained a more rounded shape (Figure 3C, left panel). This morphological phenotype was recapitulated by the shape factor that defines cell circularity, which was significantly lower in adherent cells compared to non-adherent cells with a rounder shape (Figure 3C, right panel).

To explore a possible link between BTK and β-catenin in the regulation of the cell morphology, we transfected a plasmid carrying the sequence of the human BTK gene in HEK293T cells, which do not express BTK at detectable levels but do express endogenous β-catenin, as observed via Western blotting (Figure 3D). BTK was activated in these cells, as evidenced by the signal of the phosphorylated form at Y223 (Appendix A). Under basal conditions, BTK may not affect the stability of β-catenin, as its level did not change after BTK transfection in HEK293T cells (Figure 3D). To evaluate the impact of BTK on the cell morphology, a plasmid containing human BTK associated with an IRES-GFP cassette was used to select for transfected GFP-positive cells via immunofluorescence (Figure 3E, left panel). The GFP distribution was used to trace the cell contour and calculate the shape factor, which was significantly lower in BTK-transfected cells than in cells transfected with a GFP control plasmid (Figure 3E, right panel).

Overall, these data suggest a collaboration between BTK and β-catenin in shaping CLL cells in co-culture with stromal cells.

### 2.4. β-Catenin Stabilization Is Dependent on BTK

We have previously observed that in MCL, another lymphoproliferative disorder characterized by a strong dependence on the microenvironment, the stabilization of β-catenin upon BCR stimulation is associated with BTK activity [17]. Accordingly, the pretreatment of primary CLL B cells with ibrutinib, a potent inhibitor of BTK, prevented the stabilization of β-catenin, indicating that in the presence of stromal cells, BTK is involved in β-catenin stabilization (Figure 4A). However, when we examined the phosphorylation status of BTK (Y223) in CLL B cells at different incubation times with conditioned medium, we observed a slight increase in the pBTK MFI for some patients, although it did not reach significance for the entire cohort tested (Figure 4B and Appendix A). The analysis of pBTK-positive cells showed that in the majority of samples tested, the percentage of pBTK-positive cells was already quite high and a further increase was not achievable (Appendix A). In addition, the immunoprecipitation of β-catenin from the CLL cell extract after 1 h of co-culture with stromal cells showed that β-catenin and BTK belong to the same molecular complex. Interestingly, no signal for BTK was detected in the immune precipitate from CLL cells in the absence of stromal cells (Figure 4C). To further investigate which soluble factor might be involved in β-catenin stabilization in CLL cells, we incubated CLL cells with CXCL12, a cytokine secreted by stromal cells and known to signal via BTK [24,29]. Through Western blotting, we observed that recombinant CXCL12 increased the β-catenin levels after one hour of treatment together with pERK, while no significant increase in pBTK was observed at this time point (Figure 4D). Interestingly, we observed that the CXCL12 transcript was significantly induced in HS-5 cells by the co-culture with CLL cells (Figure 4E). However, when assayed using multiarray technology, CXCL12 is greatly reduced in the co-culture supernatant, likely because it is consumed by CLL cells (Figure 4F). This result suggests that CLL cells educate stromal cells to produce CXCL12, which contributes to β-catenin stabilization in CLL cells via BTK.

### 2.5. β-Catenin Stabilization Induces Tumor Cell Adhesion to Stromal Cells

In the canonical Wnt signaling pathway, β-catenin stabilization is associated with a decrease in GSK-3β-dependent triple phosphorylation at the residues S33, S37 and T41. Accordingly, in our co-culture model, the increased stability of β-catenin observed in primary CLL cells in the presence of stromal cells corresponded to a decrease in the signal for its triple phosphorylation, once again confirming the activity of the proteasome in regulating β-catenin levels in CLL cells (Figure 5A). Furthermore, the overexpression of wild-type β-catenin did not lead to an increase in the recombinant protein, likely because it was subjected to the same post-translational regulation as the endogenous one and was constantly degraded (Figure 5B). To bypass this regulation, we constructed a mutant β-catenin (S45A) that could not be phosphorylated by either CKI or GSK-3β, and which after evading proteasomal degradation accumulated in cells, as shown in Mec-1 transfected with this mutant (Figure 5B,C). The involvement of CXCL12 in the stabilization of β-catenin led us to ask whether a higher level of β-catenin leads to increased migration towards this chemokine. The transfection of the stable form of β-catenin did not modify the migratory capacity towards CXCL12 nor stromal cells (Figure 5D). In these migration experiments, HS-5 cells were unable to induce cell migration, consistent with hMSCs having an inhibitory effect on B cell migration [30]. Overall, stabilized β-catenin did not confer a migratory phenotype to CLL cells. However, S45A-β-catenin-transfected Hg-3 cells adhere more strongly to the stromal cells, as we recovered a higher proportion of S45A-β-catenin-transfected cells when we collected HS-5 via trypsinization after removing the non-adherent cells (Figure 5E).

Since β-catenin’s stability and cellular distribution are also regulated by other phosphorylation sites, we observed that the mutant S45A-β-catenin was phosphorylated at Y654, which is known to weaken the interaction with cadherins and favor β-catenin’s nuclear accumulation [31] (Figure 5F). Interestingly, the ibrutinib treatment decreased the signal for phospho-Y654, indicating that BTK was responsible for this phosphorylation, as previously suggested in other models [31].

The nuclear enrichment of β-catenin observed via imaging cytometry in non-adherent CLL cells (Figure 2E), supported by the phosphorylation of Y654 detected on S45A-β-catenin (Figure 5F), prompted us to verify the localization of the S45A-β-catenin mutant and its transcriptional activity. Indeed, cell fractionation coupled with Western blotting showed that S45A-β-catenin is present both in the cytoplasm and the nucleus of transfected Mec-1 cells (Figure 5G). However, the transcription of Axin-2, a Wnt/β-catenin-regulated gene in most cell types, remained unchanged regardless of the presence of S45A-β-catenin-transfected cells or the co-culture with HS-5 cells (Appendix A). The lack of induction of Axin-2 was recapitulated in primary B cells, although the co-culture with stromal cells tended to reduce the transcription of this gene (Appendix A). We tested the transcriptional induction of other genes, such as cyclin D1 and c-Myc, Dll 1 and Hes 1, known to be regulated by transcription factors that are partners of β-catenin, such as TCF, Notch and NF-κB. Of all the genes tested, only cyclin D1 was induced by S45A-β-catenin but only in the presence of stromal cells (Figure 5H, left panel; Appendix A). This induction was also observed in primary cells in overnight co-cultures with HS-5 (Figure 5H, right panel). This result implies that in non-adherent CLL cells, β-catenin might possess transcriptional activity to induce cell-cycle-regulatory genes and that this activity is regulated by the microenvironment.

## 3. Discussion

β-Catenin is at the core of the canonical Wnt signaling pathway, where it translocates into the nucleus to participate in the transcription of survival genes, including Axin2 and cyclin D1 [8]. In addition, β-catenin also has an important structural role in the cell membrane to stabilize cell–cell contacts in cooperation with cadherins. There is no consensus on the interplay between the two pools of cellular β-catenin and the regulation of the two functions [19]. In some cases, the membrane β-catenin appears to reduce the amount of β-catenin available for the translocation into the nucleus, and in others the two pools do not appear to compete [20,32]. In CLL cells, the Wnt signaling pathway has received much attention because Wnt ligands are present in the microenvironment and are also secreted by CLL cells [33]. Furthermore, mutations in Wnt signaling effectors increase cell survival and resistance to apoptosis in CLL cells [6].

In this study, we focused our attention on the role of β-catenin in regulating adhesion and the relationship that may occur between adhesion and survival. When co-cultured with bone marrow stromal cells (HS-5) to mimic the microenvironment, CLL cells divide into two fractions: one pool of cells adheres to the stromal component and one pool remains in suspension. These two pools are dynamic, as the cells in suspension retain their ability to adhere when transferred to a new feeder layer of stromal cells. In turn, the adhesion is not permanent and adherent cells detach from the stromal cells. In non-adherent cells, β-catenin is rapidly stabilized by soluble factors and translocates to the nucleus to induce the transcription of cyclin D1, a target gene of the Wnt, NF-κB or STAT3 signaling pathways [34]. This increase in cyclin D1 via co-culture and β-catenin stabilization contributes to the enhanced survival of CLL cells in the presence of stromal cells. The amount of total β-catenin in the adherent CLL cells is low, suggesting that in adherent CLL cells, β-catenin is degraded in the destruction complex, as it occurs in the absence of Wnt or other stabilizing stimuli. In adherent CLL cells to the stromal cells, β-catenin rather contributes to stromal cell adhesion, as the overexpression of β-catenin increases the proportion of CLL cells that are recovered with the stromal component. In addition, β-catenin localizes in the cell membrane with polarization towards the contact zone with stromal cells. In addition, these CLL cells show ruffles as a sign of the reorganization of adhesion structures. Various observations implicate B cell signaling pathways in the functional distribution of β-catenin. First, β-catenin interacts with BTK in CLL cells in the presence of stromal cells. Second, our group has previously published the interaction between Vav1, a GTPase exchange factor (GEF) for the Rho family of GTPases expressed in hematopoietic cells, and β-catenin to affect its phosphorylation status, cellular distribution and cell morphology in a manner similar to that reported for BTK in this work [35]. Third, β-catenin interacts with Syk, which is responsible for its phosphorylation at Y142 and Y654, as we also observed for BTK in our model [36]. All of these elements come together into a model in which the B cell signaling complexes, regulated either by chemokines or stromal factors, affect the β-catenin dynamics, which ultimately tunes the balance between adhesion and transcriptional activities in these cells. The presence of a soluble factor such as CXCL12, which binds to its cognate receptor CXCR4 and activates BTK, leads to an increase in β-catenin, inducing its transcriptional activity but also its localization in the cell membrane. Interestingly, CXCL12 transcription is induced in stromal cells via a co-culture with tumor cells as part of the dialogue that exists between the two cell types. The influence on stromal cells by tumor cells allows the secretion of molecules such as CXCL12, which in turn signals to the tumor cells to increase their migration and adhesion through β-catenin and other adhesion molecules such as integrin, as previously described [24]. Membrane β-catenin favors the interaction of CLL cells with stromal cells, most likely through the interaction with cadherins, which have been reported to be enhanced via co-culture with stromal cells [22]. Using flow cytometry, we were unable to distinguish two distinct populations based on their β-catenin contents. Rather, the distribution of β-catenin labeling seems to indicate variable amounts in CLL cells. Thus, it is tempting to suggest that the adhesion of CLL cells to the stroma is a transient behavior linked to the fluctuations of β-catenin and other proteins and dictated by the stimuli that the cells receive from the microenvironment. The increase in β-catenin is important for cell adhesion to the stromal component, as ibrutinib treatment, which is opposed to β-catenin stabilization, is known to disrupt CLL cell adhesion to the cells of the microenvironment [37]. It is interesting to underline that ibrutinib does not affect the membrane expression of the integrin VLA-4 involved in CLL cell adhesion and regulated by BCR signaling [38]. Once the CLL cells adhere to the stromal component, the destruction complex could be reactivated by a negative feedback driven by mechanotransduction, leading to a decrease in cytosolic β-catenin but leaving intact the membrane pool. Other changes must occur to trigger the release of the CLL cells from the stromal cells. The phosphorylation of β-catenin in tyrosine 654 decreases its affinity for cadherins, thereby enhancing its transcriptional activity but potentially reducing its adhesion capacity by removing β-catenin from the membrane. BTK is responsible for Y654 phosphorylation in our model, thereby making this kinase an element that might tune the amount, localization and functions of β-catenin.

This work defines a new axis comprising BTK and β-catenin, which partakes in the cross-talk between tumor cells and the microenvironment and that can be targeted by ibrutinib. In addition, β-catenin is an important effector with functions in cell adhesion and transcription that intervenes in the balance between adhesion and signal transduction in response to soluble factors and mechanical cues. It is important to enrich the model of communication of CLL cells with their microenvironment because although therapeutic strategies targeting this communication are successful, resistance to treatment can develop. Therefore, finding a target to bypass this resistance will allow the effective control of a disease that is still incurable.

## 4. Materials and Methods

### 4.1. Cell Cultures

The primary CLL B cells were collected from patients after receiving informed consent during routine workup at the Avicenne Hospital (APHP, Bobigny, France). All patients gave written consent, validated by the Ethics Committee of the Avicenne Hospital in accordance with the Declaration of Helsinki. The information on the patient samples used in this manuscript is summarized in Appendix A. The primary CLL B cells were isolated from fresh blood samples using the Pan B Cell Isolation Kit (Miltenyi Biotech, Teterow, Germany) according to the manufacturer’s instructions. The isolated B cells were maintained in RPMI medium supplemented with 10% fetal calf serum (FCS), penicillin/streptomycin and glutamine and, cultured at 37 °C with 5% CO_2_ or stored frozen in liquid nitrogen in a medium consisting of 10% DMSO in FCS.

The Mec-1 and Hg-3 cells were purchased from DSMZ (Berlin, Germany), and these cell lines were cultured as described for the primary tumor B cells. The HEK293T cells were cultured in DMEM supplemented with 10% FCS, penicillin/streptomycin and glutamine and maintained at 37 °C with 5% CO_2_.

For the co-culture experiments, the ratio of HS-5 to B cells was 1:5 unless otherwise indicated. The stromal cells were plated 24 h prior to the co-culture. The ibrutinib (Selleck Chem, Cologne, Germany) was used at 100 nM. The primary CLL cells were pretreated with ibrutinib for one hour prior to the co-culture or resuspension in conditioned medium. The conditioned medium was obtained from a confluent plate of HS-5 cells that were plated 24 h before the experiment.

### 4.2. Western Blotting, Cell Fractionation and Immunoprecipitation

For the Western blots, the cells were lysed in a lysis buffer containing 50 mM Tris-HCl pH 7.4, 150 mM NaCl, 1% NP-40, 5 mM EDTA and 10% glycerol, supplemented with protease and phosphatase inhibitors (leupeptin, pepstatin, aprotinin, PMSF, NaF, sodium orthovanadate). The cell lysates were sonicated with 5 cycles of 30 sec on and off with high-amplitude pulses. After clarification by centrifugation, the protein extracts were assayed using the Bicinchoninic Acid Assay (BCA) (Pierce, Waltham, WA, USA) according to the manufacturer’s instructions. Here, 30 μg of total proteins was added of Laemmli buffer and boiled before loading onto SDS-PAGE gels. After the protein transfer, the membranes were blocked with 5% skimmed milk or BSA in TBS 0.1% Tween-20 prior to overnight incubation with primary antibodies in 2.5% skimmed milk or BSA in TBST at 4 °C. The following antibodies were used in all experiments: mouse monoclonal anti β-catenin (BD Biosciences, Le Pont de Claix, France)) (Clone 14/Beta-catenin, cat# 610153), dilution 1/2000; mouse monoclonal anti-β-tubulin (Sigma Aldrich, St. Louis, MO, USA) (Clone DM1A; cat#F2168), dilution 1/4000; rabbit anti-pS33, S37 and T41 β-catenin (Cell Signaling Technology, Saint Cyr l’Ecole, France; cat#9561), dilution 1/1000; rabbit anti-pY654 β-catenin (ECM Biosciences, Aurora, CO, USA), cat#4021), dilution 1/500; anti-BTK (Cell Signaling Technology; clone D3H5, cat#8547), dilution 1/1000; and anti-(pY223)phospho-BTK (Cell Signaling Technology, cat #5082), dilution 1/500. Incubation with the secondary antibodies, HRP-conjugated goat anti-mouse or anti-rabbit immunoglobulins (Bio-Rad, Marnes La Coquette, France); cat#0300-0108 and STAR124P), was performed at room temperature for 45 min in 2.5% skimmed milk or BSA in TBST. The chemiluminescence signal was detected using the Clarity ECL Western blot substrate (Bio-Rad) and images of the membrane were captured using a Gel-Doc EZ imaging system and ChemiDoc (Bio-Rad). The images were analyzed with the software ImageLab v.6.1 (Bio-Rad).

The cell fractionation was performed using the Subcellular Cell Fractionation Kit (Pierce) according to the manufacturer’s instructions, starting from 10 × 10^6^ of primary purified CLL B cells or 2.5 × 10^6^ Mec-1 cells. The total proteins of each fraction (cytoplasmic and nuclear) were extracted and the protein concentration was determined using the BCA protein assay kit (Pierce).

Twenty million cells were used for each immunoprecipitation (IP) experiment. For each IP, 1 mg of total protein extract was used. Here, 2 μg of the antibody was preincubated with magnetic beads, namely Dynabeads protein A (Life Technologies, Saint-Aubin, France), for 4 h in a wash buffer containing 50 mM Tris-HCl pH 7.4, 150 mM NaCl, 0.1% NP-40, 5 mM EDTA, 10% glycerol and 5% BSA. The lysates were incubated with the beads and an antibody overnight at 4 °C on a rotating wheel. After five washes, the beads were resuspended in Laemmli buffer prior to migration on 8% acrylamide gel for SDS-PAGE. The antibody against BTK used for immunoprecipitation was the same as that used for the Western blotting and the negative control was rabbit isotype IgG (2 μg; Diagenode, Liège, Belgium).

### 4.3. Confocal and Fluorescence Microscopy

Cells grown on glass coverslips precoated with poly-D-lysine (Sigma, St. Quentin Fallavier, France) were fixed with 4% paraformaldehyde (Electron Microscopy Sciences, Hatfield, PA, USA) and 4% sucrose (Sigma) for 20 min followed by 10 min of quenching with 50 mM NH_4_Cl (Sigma) and 5 min of permeabilization with 0.2% Triton X-100 (Sigma) at RT. Non-specific binding sites were blocked with 1% gelatin (Sigma) for 1 h. The cells were incubated for 1 h in a humidified chamber with primary antibodies diluted in the permeabilization buffer as follows: mouse anti-β-catenin (BD Transduction Laboratories, Le Pont de Claix, France; 1:50 dilution); rabbit anti-BTK (Cell Signaling Technology; 1:100 dilution). After performing gelatin rinses, the following secondary antibodies were applied for 45 min: anti-rabbit or anti-mouse Alexa Fluor 488; anti-rabbit or anti-mouse Alexa Fluor 647 (all from Life technologies, Thermo Fisher Scientific, Villebon sur Yvette, France; dilution 1:300). The cell nuclei and actin filaments were stained with DAPI and Alexa Fluor 546 phalloidin (Invitrogen, Thermo Fisher, Villebon sur Yvette, France) 0.3 µM and 45 min, respectively. To prevent any antibody leakage, an additional post-fixation step was performed with 2% paraformaldehyde and 2% sucrose for 10 min followed by a quenching step with 50 mM NH_4_Cl for 5 min. The coverslips were mounted using Prolong^TM^ glass antifade mountant (Molecular probes—Thermo Fisher Scientific, Waltham, WA, USA).

The 3D imaging was performed using a Zeiss LSM 780 confocal microscope system equipped with a Plan-Apochromat 63×/1.40 Oil DIC M27 objective (Zeiss, Rueil Malmaison, France). Fluorescence images were also acquired using a Zeiss Axio Observer D1 inverted-phase contrast fluorescence microscope using an EC Plan-Neofluar 100×/1.3 oil DIC objective (all from Zeiss). The shape factor was calculated using the plug-in available for ImageJ software v1.53. Over 200 cells were analyzed to perform the statistical analysis.

### 4.4. Flow Cytometry and ImageStream

For the analysis of the cell viability, a combination of anti-CD19 BV786 (clone SJ25C1, BD Biosciences) and anti-CD90 BUV496 (clone 5E10, BD Biosciences) was resuspended in Brilliant Blue staining buffer and added to the 50,000 cells. After 20 min of incubation at 4 °C in the dark, a washing step was performed using a binding buffer solution. The pellet was resuspended in the binding buffer solution and Annexin-V APC and 7-AAD (BD Biosciences) were added according to manufacturer’s instructions. After 15 min of incubation at room temperature in the dark, a washing step was performed using the binding buffer solution. The cells were analyzed with BDFACS Symphony A3.

For the flow cytometry experiments, the CLL B cells were labeled with anti-CD19-PE (clone HIB19, BD Biosciences) and HS-5 with anti-CD90-PE-Cy5 (clone 5E10, BD Biosciences). For the intracellular labeling of β-catenin, the Cytofix/Cytoperm Kit was used according to the manufacturer’s instructions (BD Biosciences). After fixation for 20 min, the cells were washed with the permeabilization solution and incubated with anti-β-catenin conjugated to Alexa488 or anti-β-catenin APC conjugated (clone 14, cat#562505, BD Biosciences) for 1 h before acquisition on a Canto II cytometer (BD Biosciences). A control with extracellular labeling only was used as the FMO (fluorescence minus one) to determine the positivity of the intracellular staining. For the labeling of pBTK, Alexa488 anti-pBTK (Y223) (BD Biosciences, cat#564846) was used after fixation in paraformaldehyde 2% in PBS at RT for 20 min, followed by permeabilization with a buffer containing 0.5% saponin (Sigma Aldrich, St. Louis, MO, USA) and 1% bovine serum albumin in PBS 1× for 30 min at RT. The cells were incubated with pBTK for 1 h at 4 °C.

For the analysis on the ImageStream system (MKII-Luminex, Merck, Darmstadt, Germany), the protocol was the same as for the intracellular staining, except that the DAPI (dilution 1/10,000) was added 5 min before acquisition. A statistical analysis was performed using the predefined nuclear translocation template from IDEAS software v.6.0 after the establishment of a compensation matrix.

### 4.5. Cell Transfection and Cell Adhesion Assay

HEK293T cells were plated the day before transfection to reach 70% confluence on the day of transfection. Trans-IT (Euromedex, Souffelweyersheim, France) was used to transfect the HEK293T cells with 2.5 μg of DNA with the plasmid SFVL containing the coding region of human BTK, which was a gift from Prof. S. Stilgenbauer (University of Ulm, Ulm, Germany).

The plasmid pcDNA3.1 containing the wild-type β-catenin coding region was a gift from Dr. E. Fearon (Addgene, Watertown, MA, USA; cat#16828). The S45A mutant was generated via site-directed mutagenesis. A total of 7.5 × 10^6^ Hg-3 cells were co-transfected with either 10 µg of pcDNA-S45Aβ-catenin or the control plasmid and 1 µg of pmax GFP supplied with the Amaxa kit using the Amaxa B cell line Nucleofector Kit V according to the manufacturer’s instructions with the U-14 program (Amaxa Lonza, Cologne, Germany). Twenty-four hours after nucleofection, 5 × 10^6^ Hg-3 cells were co-cultured for 2 h with 1 × 10^6^ of HS-5 cells, which were preseeded 24 h prior to the co-culture in 6-well plates. The non-adherent Hg-3 cells were removed and the adherent Hg-3 and HS-5 cells were carefully washed four times with DPBS (Thermo Fisher Scientific, Waltham, WA, USA). The HS-5 and adherent Hg-3 cells were trypsinized and analyzed via flow cytometry. The HS-5 cells were labeled with anti-CD90-PE antibody, whereas the Hg-3 cells were labeled with anti-CD19-PE-Cy7 antibody. The proportions of transfected and adherent Hg-3 to HS-5 cells were determined by selecting CD19^+^GFP^+^ cells.

For the adhesion–detachment experiments, 7.5 × 10^5^ HS-5 cells were plated in a 6-well plate 24 h before the adhesion experiment. Hg-3 cells were labeled with Deep Red Dye solution (Thermo Fisher Scientific, Waltham, WA, USA) and added to the HS-5 feeder layer at a HS-5/Hg-3 ratio of 1:5. The cells were allowed to adhere for 2 h prior to the removal of the non-adherent cells. The same number of unlabeled Hg-3 cells were added to the same feeder layer and allowed to adhere for a further 2 h before counting the labeled and unlabeled Hg-3 cells in the adherent and non-adherent fractions via flow cytometry. Anti-CD90 PE and anti-CD19 APC-Cy7 antibodies were used to label the HS-5 cells and Hg-3, respectively.

### 4.6. Cell Migration Assay

Cell migration experiments were performed using 12-well plates supplied with 8 µm inserts (Corning, Boulogne-Billancourt, France). Hg-3 cells were transfected with an empty plasmid or a plasmid containing S45A-β-catenin, as described above, 24 h before the experiment and then seeded in the upper compartment. The cells were allowed to migrate to the lower compartment for 4 h under the following conditions: RPMI supplemented with 10% FCS, CXCL12 100 ng/µL in complete medium or HS-5 cells seeded 24 h before the experiment. The ratio between the Hg-3 and HS-5 cells was 5:1. The same number of transfected Hg-3 cells were seeded in a well without a transwell and used to count the total number of cells to be used as the denominator in the ratio to calculate the percentage of migrated cells.

### 4.7. Quantification of Cytokine Secretion

CXCL12 was quantified in the supernatants of HS-5 cells when cultured alone or co-cultured with CLL B cells for 24 h using U-Plex assays (MSD, Rockville, MD, USA) according to the manufacturer’s protocol.

### 4.8. RT-qPCR

The total RNA was purified using the RNeasy Mini kit (Qiagen, Les Ulis, France) and at least 500 ng of total RNA was retro-transcribed using iSCRIPT (Bio-Rad) according to the manufacturer’s instructions. The amplicons were amplified and quantified using SYBR Green with the following primer sets: β-catenin forward 5′-CGTGCACATCAGGATACCCA-3′ and reverse 5′-ATTTCTTCCATGCGGACCCC-3′; c-myc forward 5′-CTCCGTCCTCGGATTCTCTG-3′ and reverse 5′-CTTGTTCCTCCTCAGAGTCGC-3′; Dll1 forward 5′-AGAAGGATGAGTGCGTCAT-3′ and reverse 5′-TTTAAGAGAAACGGGAGTCTTG-3′; -Hes1 forward 5′-GAAGAAAGATAGCTCGCGG-3′ and reverse 5′-TTCCGGAGGTGCTTCAC-3′; cyclin D1 forward 5′-ACCTGGATGCTGGAGGTCT-3′ and reverse 5′-GCTCTTTTTCACGGGCTCCA-3′; β-2microglubulin used as housekeeping gene forward 5′-CTCCGTGGCCTTAGCTGTG-3′ and reverse 5′-TTTGGAGTACGCTGGATAGCCT-3′; Axin-2 forward 5′-GGACAGGAATCATTCGGCCA-3′ and Axin-2 reverse 5′-ACCTGCCAGTTTCTTTGGCT-3′.

The qPCR was performed on a StepOnePlus^TM^ system (Thermo Fisher Scientific, Waltham, WA, USA) and the data were analyzed using StepOne software v.2.3 and GraphPad 6.0c for the statistics.

The expression of CXCL12 and cyclin D1 was analyzed using TaqMan Gene Expression Assay probes (Thermo Fisher Scientific, Waltham, WA, USA) (CXC12 probe Hs03676656_mH; cyclin D1 probe Hs00765553_m1) on a 7500 Real-Time PCR system (Life Technologies, Saint-Aubin, France). The target gene expression was normalized to the mean Ct values of the housekeeping gene GAPDH (Hs02786624_g1). The normalized ratio is expressed as 2^−ΔCt^. All tests were performed in duplicate.

## Figures and Tables

**Figure 1 ijms-24-17623-f001:**
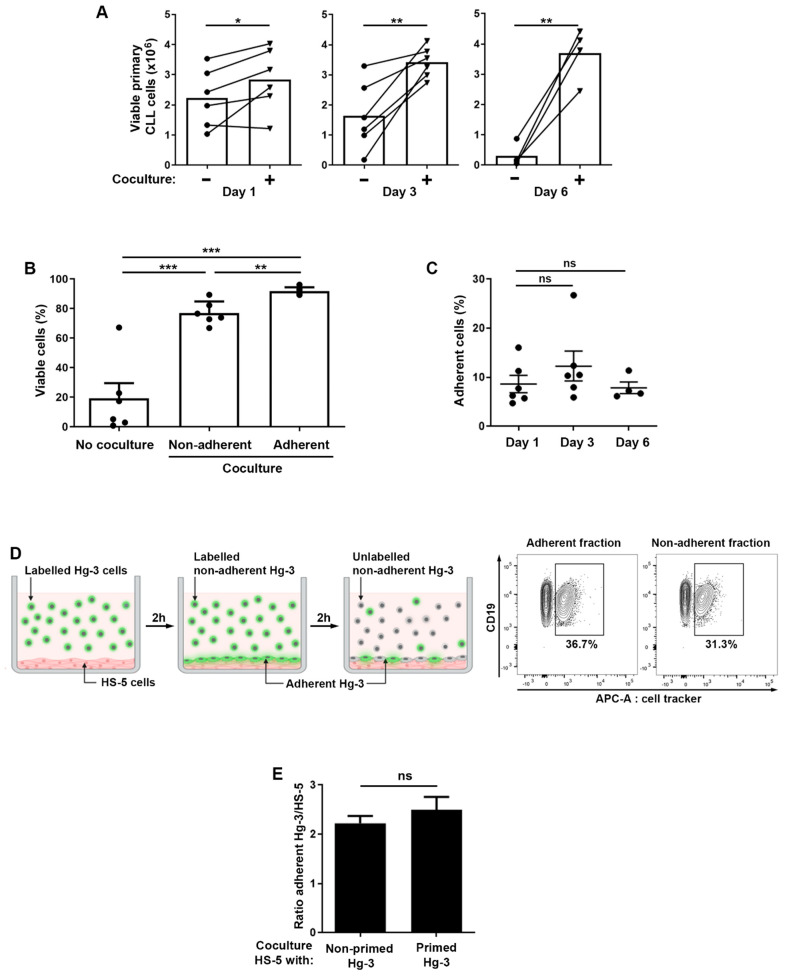
Adhesion and survival analyses of CLL cells cultured with stromal HS-5 cells. (**A**) An analysis of cell viability based on AnnexinV/7-AAD labeling of primary CLL cells in the presence or not of HS-5 cells for up to 6 days at a ratio of 10:1, respectively. Viable cells are those of AnnexinV-/7-AAD- (*n* = 6 for days 1 and 3; *n* = 4 for day 6). (**B**) Percentage of viable primary CLL cells (AnnexinV-/7-AAD-) cultured in the presence or not of HS-5 cells and comparison of survival in adherent versus non-adherent cells (*n* = 6) at day 6. The data represent the mean ± SEM. (**C**) Flow cytometry was used to assess the adhesion of primary CLL cells to stromal cells (HS-5) in a co-culture system. CLL cells and HS-5 cells were co-cultured for 1, 3 and 6 days. After the removal of the non-adherent cells, the HS-5 and adherent CLL cells were collected via trypsinization. The percentage of CLL cells recovered in the stromal fraction relative to the total number of CLL cells is shown on the graph (*n* = 6 for days 1 and 3; *n* = 4 for day 6). (**D**) Cartoon depicting the strategy used to determine the dynamics of CLL cell adhesion to stromal cells. On the right, a representative cytometry plot shows the gating strategy and the analysis of the Cell Tracker-APC-positive Hg-3 cells in the adherent and non-adherent fractions. (**E**) Graph showing the difference in adhesion between Hg-3 cells (non-primed Hg-3) and those previously exposed to HS-5 cells for 2 h in co-culture (primed Hg-3). Data are expressed as the ratio of CD19-positive cells (Hg-3) to CD90-positive cells (HS-5) (*n* = 3). Note: ns: not significant; * *p* < 0.05, ** *p* < 0.01, *** *p* < 0.001, Student’s *t*-test.

**Figure 2 ijms-24-17623-f002:**
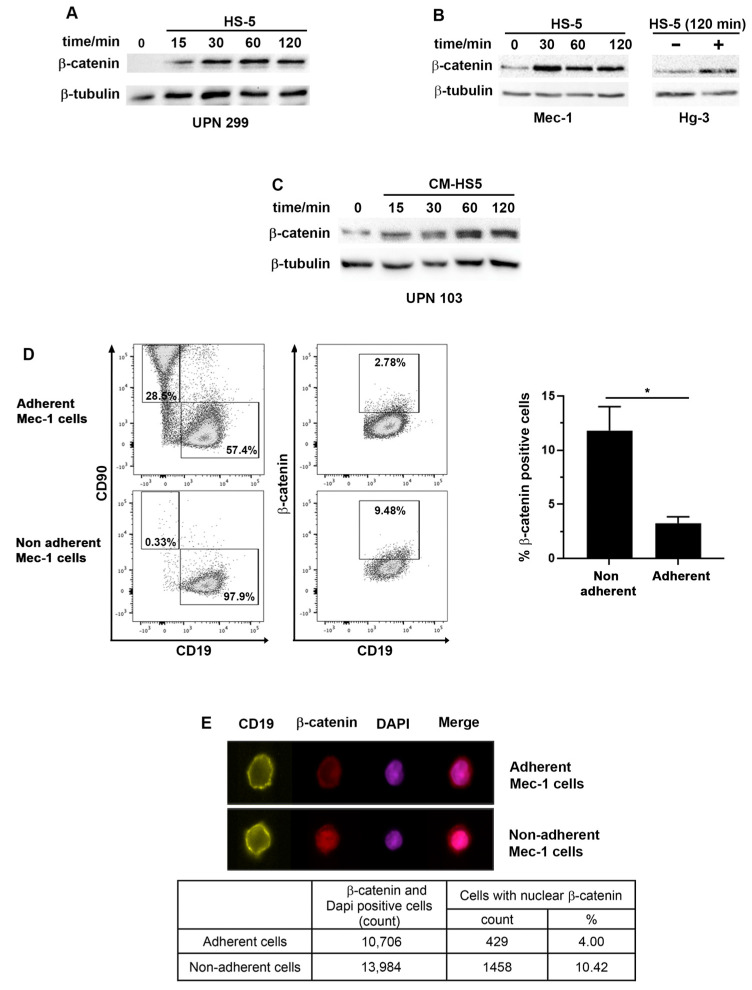
The levels of β-catenin expression and localization are different in adherent versus non-adherent CLL B cells. (**A**) A Western blot analysis of β-catenin from CLL B cells (UPN 299) co-cultured with the human stromal cell line HS-5 for different incubation times. Here, β-tubulin is used as a loading control. Images are representative of at least three independent experiments. (**B**) A Western blot analysis of β-catenin stabilization in Mec-1 and Hg-3 cells as described for primary cells. Only the 120 min time point is shown for Hg-3 cells. (**C**) A Western blot analysis of β-catenin from primary CLL cells (UPN 103) cultured in conditioned medium issued from a 24 h culture of HS-5. (**D**) Gating strategy used to detect β-catenin in co-cultures of Mec-1 cells (CD19^+^) and stromal cells (CD90^+^). Percentages of cells in each gate (rectangle) are indicated. The histogram on the right shows the statistical analysis of β-catenin-positive cells in adherent versus non-adherent cells performed on 3 independent experiments. Note: * *p* < 0.05, Student’s *t*-test. (**E**) Image and flow cytometric analyses of CLL cells in co-culture with HS-5 cells (Magnification 40×). After 2 h of co-culture, the non-adherent cells were removed and labeled separately from the adherent fraction obtained via trypsinization. The cells were labeled with CD19-PE to identify CLL cells. After fixation and permeabilization, β-catenin was labeled with an APC-conjugated antibody. The nuclei were labeled with DAPI. CD19-positive cells were analyzed for the β-catenin signal and DAPI. The table shows the number of cells and the percentage of cells with overlapping β-catenin and DAPI signals for adherent and non-adherent cells.

**Figure 3 ijms-24-17623-f003:**
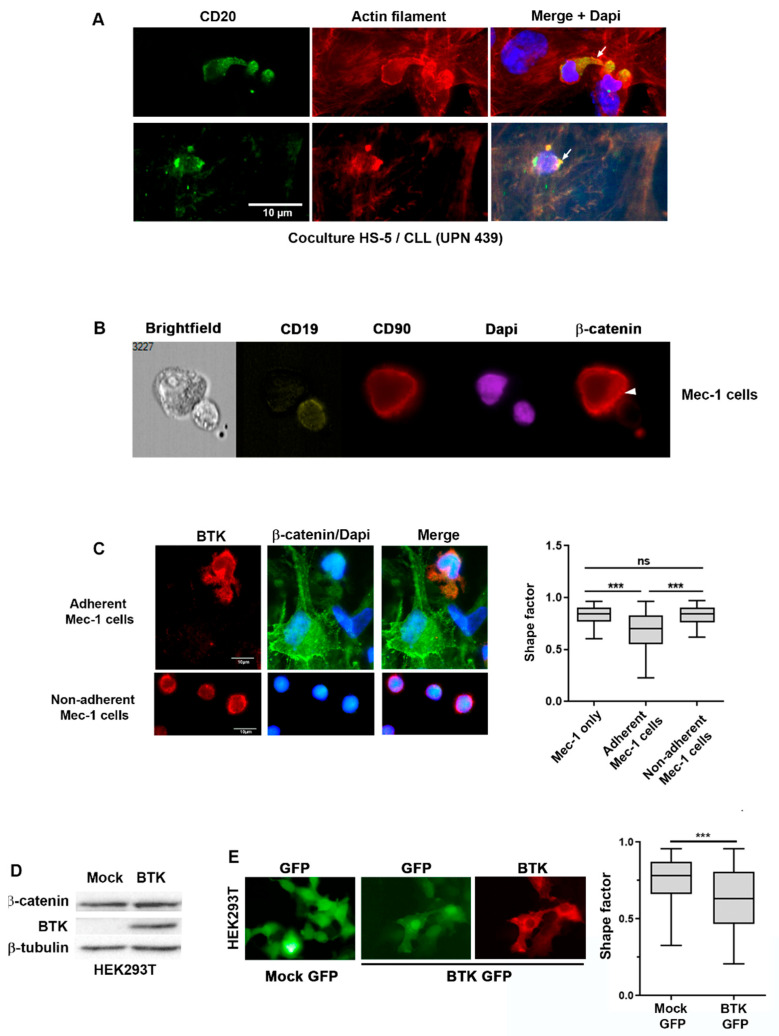
β-Catenin and BTK cooperate to influence the CLL cell morphology in co-culture with stromal cells. (**A**) Immunofluorescence of co-culture of stromal cells (HS-5) and CLL cells labeled with an anti-CD20 antibody (green). Phalloidin labels the actin cytoskeleton (red). The white arrow in the merged image points to the protrusions formed by CLL cells in contact with stromal cells. The images are all issued from the same sample (UPN 439) but represent two different fields from the same coverslip. CLL cells from three different patients were analyzed in this experiment. (**B**) Image Stream analysis of doublets of Mec-1/HS-5 cells showing the localization of β-catenin (white arrow) in the two cell types. (**C**) Immunofluorescence images showing BTK and β-catenin in Mec-1 cells that adhere (top images) and do not adhere (bottom images) to stromal HS-5 cells. BTK and β-catenin are labeled by antibodies and are shown in red and green, respectively. The panel on the right shows the shape factor. An average of 200 cells were used to calculate and compare the shape factors between primary CLL cells cultured alone (Mec-1 only) and in the non-adherent and adherent fractions. (**D**) Immunoblotting of protein extracts from HEK293T cells transfected with a plasmid carrying the human BTK gene or with an empty plasmid, using anti-BTK and anti-β-catenin antibodies. β-Tubulin was used as a loading control. (**E**) Microscopy images of HEK293T cells transfected with human BTK-IRES-GFP or a GFP-containing vector (mock) (Magnification 40×). A shape factor analysis compares the cell morphologies in these two conditions. Note: ns: not significant; *** *p* < 0.001, Student’s *t*-test.

**Figure 4 ijms-24-17623-f004:**
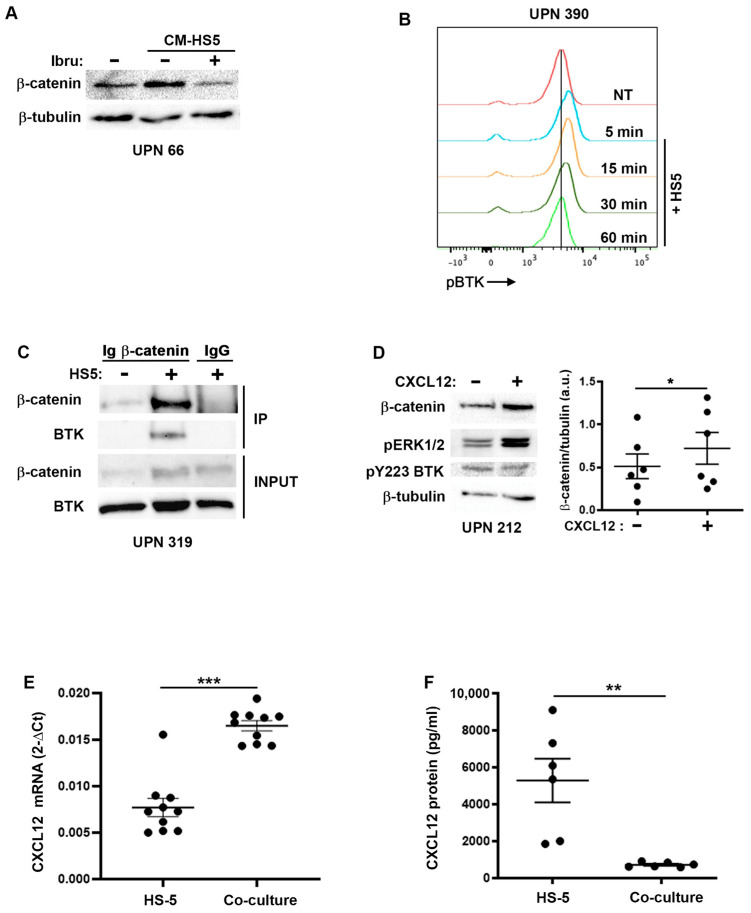
BTK is involved in β-catenin stabilization in CLL cells co-cultured with HS-5. (**A**) Primary CLL cells (UPN 66) were pretreated or not with ibrutinib (100 nM) for 1 h prior to incubation with conditioned medium issued from a 24 h culture of stromal cells. The β-catenin and β-tubulin were detected via Western blotting. (**B**) Histogram representing the analysis via phosphoflow of pBTK (pY223). (**C**) Co-immunoprecipitation of β-catenin and BTK from purified primary CLL B cells (UPN 319) co-cultured with or without HS-5 stromal cells for one hour. The input represents 10% of the total lysate and was set aside prior to immunoprecipitation. Mouse IgG was used as a negative control for the immunoprecipitation. (**D**) Representative immunoblot (UPN 212) and quantification of β-catenin levels after treatment with CXCL12 (100 ng/mL) for 1 h from purified primary CLL cells (*n* = 6). The pERK1/2 was used as a positive control for the stimulation, pY223 BTK as an activated protein marker and β-tubulin as a loading control. (**E**) RT-qPCR to detect CXCL12 transcript in HS-5 after 24-h co-culture with primary CLL cells (*n* = 10). GAPDH was used as a housekeeping gene in this experiment. (**F**) Multiarray technology was used to quantify CXCL12 in the supernatants of 24-h co-cultures of primary CLL cells and HS-5 cells (*n* = 6). Note: * *p* < 0.05, ** *p* < 0.01, *** *p* < 0.001, Student’s *t*-test.

**Figure 5 ijms-24-17623-f005:**
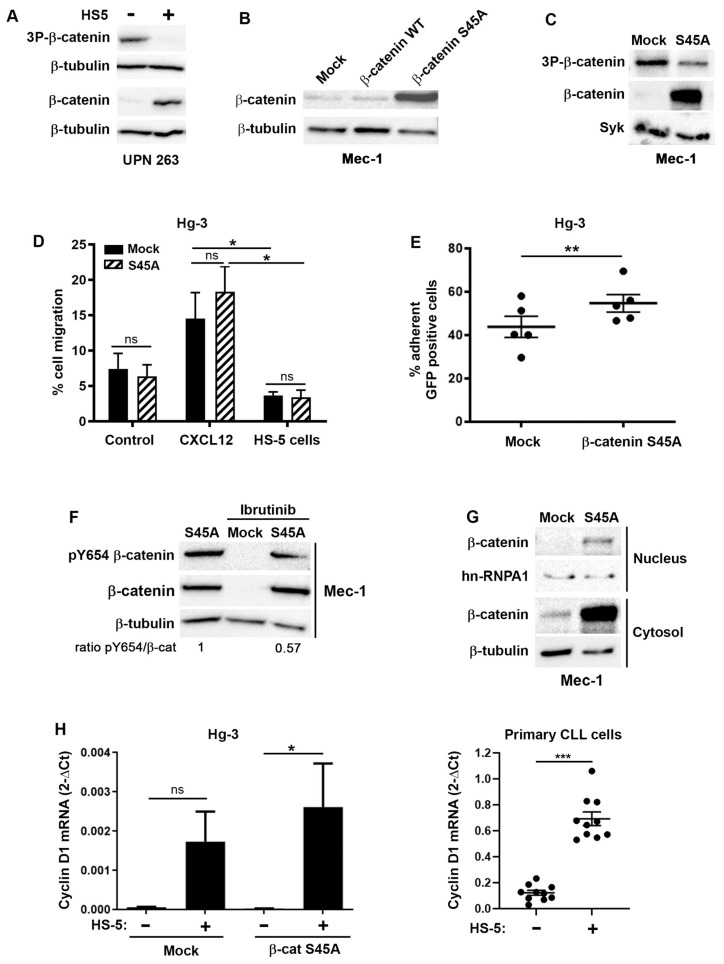
Functional involvement of β-catenin in CLL cells. (**A**) A Western blot analysis of the phosphorylation status of β-catenin after co-culture or not with HS-5 stromal cells for 1 h. Whole-cell lysates of primary purified CLL cells (UPN 263) were probed with the antibodies directed against the triple phosphorylation sites S33, S37 and T41 (3P-β-catenin); total β-catenin; or β-tubulin. The cell lysates were loaded onto two different membranes probed with anti-3P-β-catenin and total β-catenin, respectively. The loading controls (β-tubulin) of the two membranes are shown. (**B**) A Western blot analysis of the whole-cell lysates of Mec-1 cells transfected with an empty plasmid (mock) or a plasmid carrying the wild-type human β-catenin gene or the mutant form (S45A). (**C**) A Western blot analysis of the whole-cell extracts of Mec-1 cells transfected with an empty plasmid (mock) or a plasmid carrying the mutated form of β-catenin (S45A). The immunoblot was probed with anti-β-catenin or 3P-β-catenin antibodies; Syk was used as a loading control. (**D**) Migration experiments performed in a transwell (8 μm diameter pores) with Hg-3 cells transfected with an empty plasmid (mock) or the S45A-β-catenin. Transfected Hg-3 cells were seeded on the upper chamber and allowed to migrate for 4 h towards the lower chamber containing RPMI, CXCL12 or HS-5 cells. The graph shows the percentage of migrated cells over the total number of seeded cells from 3 independent experiments. (**E**) Graph of the flow cytometry data used to calculate the percentage of S45A-β-catenin versus empty plasmid (mock) transfected Hg-3 cells adhering to HS-5 after 1 h of co-culture. GFP was used to count transfected Hg-3 cells as described in the Section 4 (*n* = 5). (**F**) An analysis of the phosphorylation of β-catenin at Y654 in Mec-1 cells transfected with the S45A mutant or an empty plasmid (mock). Where indicated, the cells were pretreated or not with ibrutinib (100 nM) for 1 h. The Western blot is representative of 3 independent experiments. (**G**) A cell fractionation protocol followed by immunoblotting was used to detect β-catenin with specific antibodies in the cytosol and nucleus of Hg-3 cells transfected with the plasmid carrying S45A-β-catenin or an empty plasmid (mock). β-Tubulin and hnRNP A1 were used as loading controls for the cytosolic and nuclear fractions, respectively. (**H**) RT-qPCR for cyclin D1 performed on Hg-3 (left panel) and primary CLL cells (right panel). Hg-3 cells were transfected with an empty plasmid (mock) or with S45A-β-catenin and cultured in the presence or absence of HS-5 cells for 4 h. The graphs represent the average data from 3 independent experiments. Primary CLL cells were cultured with HS-5 cells for 24 h before the RNA extraction (*n* = 10). Delta Ct values are calculated as subtractions of cyclin D1 C_t_ and of the housekeeping gene, β2 microglobulin. Note: ns: not significant, * *p* < 0.05, ** *p* < 0.01, *** *p* < 0.001, Student’s *t*-test.

## Data Availability

All data generated and analyzed in this work are available from the corresponding authors upon request.

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
