# Peer review of "Identification of the Axis β-Catenin–BTK in the Dynamic Adhesion of Chronic Lymphocytic Leukemia Cells to Their Microenvironment"

_ijms, 2023, doi:10.3390/ijms242417623_

Round 1

Reviewer 1 Report

Comments and Suggestions for Authors

In manuscript IJMS-2735807, Mihoub et al. study the effects of the interaction of CLL with stromal cells in vitro on b-catenin-mediated adhesion/transcription balance. The study is interesting and appears to be properly conducted and well written.

However, as a major point of critique, the title suggests that transcriptional changes upon cell-cell interaction were investigated, but these were not sufficiently addressed (qPCR of only a few targets). A global approach (e.g. RNA-seq) should be conducted with CLL cell lines and primary samples cultured on versus without feeders and analyzed bioinformatically (GSEA etc.). Or the title should be adapted.

Minor points:

Fig. 1B: analysis on which day? Please generally disclose what the error bars mean (standard deviations?).

According to Fig. 2B caption Hg-3 were incubated for 120 min, according to the legend 1h. What is correct?

In Fig. 2D the CD90 signal of adherent Mec-1 cells spreads much more than non-adherent. Is this due to compensation artefacts or altered cell shape (are FSC and/or SSC also different? If yes, this would be in line with the subsequent findings). Furthermore, I would term the gated b-catenin fraction “high” instead of “positive” since the other cells express low basal levels as well.

Figs. 3B + S3A: as far as I can judge from the presented microphotographs only the stroma cells appear to be polarized for b-catenin. Is the resolution high enough to resolve both contacting membrane areas sufficiently?

Fig. 5H: please disclose p-values for the differences between Hg-3 – and + HS-5. Both (Mock and S45A) appear to be significant.

For reproducibility reasons, the authors should disclose more methodological and reagents details, e.g. precise antibody identities (RRIDs) and dilution factors, Axin-2 primers etc. Furthermore, legends for the supplementary figures are missing.

The authors used chemiluminescent detection and quantification of Western blot band signals. Such a procedure is only valid if the signals (targets, housekeeping protein) lie within the linear range of detection. Has this been confirmed e.g. by dilution series? Is the housekeeping protein (b-tubulin) stably expressed in this setting?

The authors should discuss how far their results obtained with a rather artificial model (feeder-independent cell lines Hg-3 or Mec-1 grown on HS-5 cells) may be representative for the in vivo situation in patients (stroma-dependent CLL).

Comments on the Quality of English Language

English seems to be okay.

Reviewer 2 Report

Comments and Suggestions for Authors

Mihoub and colleagues study roles of canonic wnt signalling in CLL cell interactions with the tumor microenvironment. These are timely studies as evidence is accumulating that adhesion to stroma protects, and de-adhesion from stroma sensitizes tumor cells to chemotherapeutic agents. Particularly attractive is the observation that adhesion to stroma is not a property distinguishing two different CLL cell types, but rather a seemingly stochastic event, and that adhesion vs. non-adhesion are transient, dynamic events. Accumulation of dead cells in the non-adherent fraction (Fig. 1B) may not necessarily indicate that non-adhesion kills cells, since adhesion is an active process, and dead cells should be released from stroma and “contaminate” non-adherent live cells. In Fig. 2D, ß-catenin staining does not really look positive vs. negative, but the staining intensity of the entire population is shifted. More accurate than “% ß-catein positive cells” would probably be a comparision of MFI. This does not change interpretation of the data – clearly non-adherent cells have a stronger ß-catenin signal.

Round 2

Reviewer 1 Report

Comments and Suggestions for Authors

The authors responded adequately to my comments. I now consider the paper suitable for publication.